# Sportspersonship orientation in the training process of young football players in different training contexts

Gema Ortega Vila[1], José Robles Rodríguez[2], Manuel Tomás Abad Robles[2]*, Enrique Ortega Toro[3], Francisco Alarcón López[4], Francisco Javier Giménez Fuentes-Guerra[2]

1 Real Madrid Foundation, Madrid, Spain, 2 Faculty of Education, Psychology and Sports Sciences, University of Huelva, Huelva, Spain, 3 Faculty of Sport Science, Regional Campus of International Excellence "Campus Mare Nostrum," University of Murcia, Murcia, Spain, 4 Faculty of Education, Department of General and Specific Didactics, University of Alicante, Alicante, Spain

* manuel.abad@dempc.uhu.es

**Data Availability Statement:** All relevant data are within the manuscript and its Supporting Information files.

## Abstract

### Background

Football plays a fundamental role in the lives of the boys and girls who play it and it can become an invaluable educational instrument. However, for introduction to football to be truly educational, it needs to be carefully planned and guided; otherwise, it may encourage negative and undesirable behaviour from a moral and ethical point of view. That way, many authors claim that a careful and deliberate initiation to football is necessary for the promotion of sportspersonship to become a reality.

### Purpose

The goal of this research is to analyse the evolution of sportspersonship orientation in the social sports football schools of Fundación Real Madrid and in association football clubs. For this purpose, the Multidimensional Sportspersonship Orientations Scale is used. The research involves 1509 male and female youth football players aged between 8 and 16 years (M = 10.9±2.2), of whom 952 belong to the social sports football schools of the Fundación Real Madrid and 547 to association football clubs.

### Results

The most outstanding results show that the players of the social sports schools of Fundación Real Madrid obtain high values in sportspersonship orientation. Moreover, these values remain high in all categories, while they show a significant decrease in players from association football clubs.

### Conclusions

The social sports model of Fundación Real Madrid can contribute to maintaining high levels of sportspersonship orientation throughout the player's training process.

**Funding:** Real Madrid Foundation. The funders had no role in study design, data collection and analysis, decision to publish, or preparation of the manuscript.

**Competing interests:** No authors have competing interests. The interest of the Real Madrid Foundation in this study was to evaluate the sports model that they develop in order to continue to improve it.

## Introduction

Football plays a fundamental role in the lives of the boys and girls who play it and it can become an invaluable educational instrument [1]. However, for introduction to football to be truly educational, it needs to be carefully planned and guided [2]; otherwise, it may encourage negative and undesirable behaviour from a moral and ethical point of view [3]. Sportspersonship encompasses behaviours that reflect the moral and character development of young people in contexts of physical activity and sports [4]. Hence, sportspersonship may include behaviours linked to social norms (such as showing respect, self-control, etc.), as well as social behaviour that shows concern for the well-being of others (e.g., cheering a team mate, helping an injured opponent, etc.) [5]. According to Wandzilak et al. [6] a sporting conduct is related to behaviour defined by a positive interaction with team mates, referees, coaches and opponents. The term sportspersonship is used to describe behaviour that reflects the moral and character development of a boy, girl or teenager in contexts of physical activity and sports, which may include behaviour related to social norms and to the well-being of others [4]. Apart from that, from a social and psychological perspective, for Vallerand et al. [7] sportspersonship comprises five aspects or factors: 1) commitment to, effort in and total involvement with their sport, 2) respect for social conventions, 3) respect for rules and referees, 4) respect for the opponent, 5) a negative approach to sports or disruptive behaviour of the player.

To create experiences that promote sportspersonship can contribute to increasing the enjoyment of boys and girls who engage in physical activity and sports, as well as fostering its inclusion as part of their lifestyle, thus preventing sports drop-out [8]. Thus, there is a need for specific training of sports coaches in relation to the development of sporting behaviour [9]. Furthermore, in order to contribute to the promotion of positive sporting attitudes, it will be necessary to move away from traditional sporting models, in which competition, results or early specialisation are prioritised [3,10].

The development and promotion of sportspersonship will probably be highly influenced by the sporting model chosen by clubs and sports schools. On the one hand, the model of federations, which is the most common practice in Spain, is formal and regulated through the corresponding sports federation (in this case, the football federation). It is a traditional model based on practice, technique, competition and the search for results. According to Sánchez [11], this model is governed by the principle of selection, so that the sporting elite becomes, by definition, a minority group and entails great sacrifice. Still, there are other models. The FRM's social sports football schools are based on a sporting and pedagogical model in which training principles are both to educate and promote values, and to teach the game itself. Since this sporting model seeks an alternative practice of sports, the competition needs to be amended and adapted to its new aims. Thus, it will become a great learning method in which the rules will be different and adapted to each age group, the maximum participation of the players will be sought, avoiding selection by level, and neither the classification nor the result will be so important. This "White Card" programme is particularly relevant in the competitive model, which rewards the good behaviour of the players in the internal competition tournament with a White Card in recognition of certain specific behaviour or of their general record of sportspersonship. The main goal of this programme is to motivate sporting behaviour among the players, who gradually internalise such honest attitude to the point where it becomes common and natural.

As can be seen in recent years, research on the development of values through sport has increased greatly, which has contributed to the study of sportspersonship [12–15]. However, research focused on analysing the development of values such as sportspersonship in different

context are scarce [16]. That is why the aim of this study was to analyse the evolution of sports-personship orientation in the social sports schools of Fundación Real Madrid (FRM) and in association football clubs.

## Methodology

### Participants

The research has involved 1509 male and female youth football players aged between 8 and 16 years ($M$ = 10.9±2.2), of whom 952 belonged to the social sports football schools of FRM and 547 to association football clubs. The participants were recruited in the season 2021/22, from 01/10/2021 to 01/05/2022. Players from the FRM schools had been training for between 1 and 12 years ($M$ = 3.1±2.1), while those from association football clubs had been training for between 1 and 13 years ($M$ = 4.9±2.7). The sample included 1480 boys (98%) and 29 girls (2%). In terms of the different categories, 485 (32.1%) belonged to the U10 category (FRM = 349; clubs = 136); 452 (30%) to the U12 category (FRM = 302; clubs = 150); 369 (24.5%) to the U14 category (FRM = 212; clubs = 157); and 203 (13.5%) to the U16 category (FRM = 99; clubs = 104).

### Instrument

The tool used was the *Multidimensional Sportspersonship Orientations Scale* [14]. This questionnaire has 21 items and is classified in two dimensions: personal and social factors. The former refer to the participants' commitment to the sport, their emotional control in dealing with mistakes, and their obedience to the coach. The latter comprise respect for social conventions, rules, opponents and the referee; fairness to all; and courtesy to other players, as well as helping others. Each item is answered on a Likert-type scale with 5 options to express the degree of agreement or disagreement with each of the different questions: (1) *I do not agree with it at all*, (2) *I hardly agree with it*, (3) *I partly agree with it*, (4) *I mostly agree with it*, y (5) *I completely agree with it*. The analysis of internal consistency (Cronbach's alpha) showed values of .830 for the entire scale. Regarding the alpha in each dimension, the following data were observed: personal factors .712 and social factors .791.

### Procedure

The research was carried our upon approval by the ethical review board of FRM, as well as by the boards of the association football clubs. Furthermore, families signed an informed consent form explaining the purpose of the study. Participation has been voluntary in all cases. They were also informed that they can leave the research at any point in time. The research is framed within the international ethical Declaration of Helsinki (2013), the recommendations of the WHO, the code of ethics, the regulations on data confidentiality, and the Spanish Organic Law 3/2018 of 5 December on Data Protection and Guarantee of Digital Rights. The general research project on different variables to study the FRM social sports schools has been approved by the Bioethics Committee of the Andalusian Regional Government (Code: 0803-N-20).

Questionnaires were distributed in accordance with the following protocol: both in the case of FRM and in the rest of the football clubs, the coordinators of the different schools and clubs held a meeting with a member responsible for the research, who explained the procedure. The questionnaire was filled just before the start of one of the training sessions. Coordinators were present at all times and read out each question in case any respondent had any doubts, in which case the coordinator provided the appropriate explanations to the players.

## Statistical analysis

A basic descriptive analysis of central tendency and dispersion was carried out for each of the dimensions. To determine whether there were differences between the two contexts (WRF vs federated clubs), the independent samples Student's $t$-test was used. Where data showed significant differences ($p < .05$), effect sizes were calculated using standardised mean differences (Cohen effect size) [17] using the following reference values: small ($d = 0.2$), medium ($d = 0.5$) and large ($d = 0.8$) effect.

To analyze whether there was interaction effect between Context (FRM vs Clubs) and Age Categories (U10, U12, U14, U16), a two-factor ANOVA (2x4) was performed. An estimated marginal means analysis was performed; simple main effects were compared by post hoc test with Bonferroni confidence interval adjustment. Effect size was calculated by partial eta squared ($\eta^2_p$, using the following reference values [17]: no effect ($\eta^2_p < .01$), small effect ($.01 \leq \eta^2_p < .06$), medium effect ($.06 \leq \eta^2_p < .14$) and large effect ($\eta^2_p < .14$). Data were calculated using SPSS statistical software, version 29.0, with a significance level of $p < .05$.

## Results

### Assessment of sportspersonship orientation and differences between the FRM schools and the association football clubs

Regarding the assessment of sportspersonship orientation, the data showed that the FRM players presented higher values than those of the federated clubs (FRM = 4.098±.579; clubs = 3.961 ±.564), detecting significant differences ($p < .001$) with a small effect size ($d = .24$). Although the scores for sportspersonship were high on both dimensions, personal factors scored higher than social factors. When comparing the results between the FRM schools and the association football clubs, it was observed that there were differences in both dimensions. The data showed significant differences with regard to personal factors, with players from association football clubs giving greater importance to these aspects, with a small effect size. In contrast, social factors were more highly valued by the FRM boys and girls, with a moderate effect size (Table 1).

### Evolution of sportspersonship by category in the FRM schools and in the association football clubs

When analysing the evolution of sportspersonship orientation in the different categories, uneven progress was observed in both sporting contexts. The ANOVA test revealed, as far as the FRM schools were concerned, that the high values obtained by the players in the U10 category also remained high in the rest of the categories. On the contrary, the values obtained by the boys and girls of the clubs showed a considerable decrease in higher categories. ANOVA showed significant differences ($p < .001$) with a large effect size ($\eta^2 = .186$) (Fig 1). The *post*

**Table 1. Descriptive data of the dimensions.**

|  | Total (n = 1509) | FRM (n = 952) | Clubs (n = 547) |  |  |
|---|---|---|---|---|---|
|  | M±SD | M±SD | M±SD | p | d |
| Personal fact. | 4.181±.602 | 4.125±.650 | 4.279±.493 | < .001 | .257 |
| Social fact. | 3.915±.730 | 4.070±.653 | 3.643±.778 | < .001 | .609 |
| Sportspersonship | 4.048±.577 | 4.098±.579 | 3.961±.564 | < .001 | .24 |

Note. M = Mean; SD = Standard deviation; p = significance; d = Cohen's d.
Grouping variable: Sporting context.

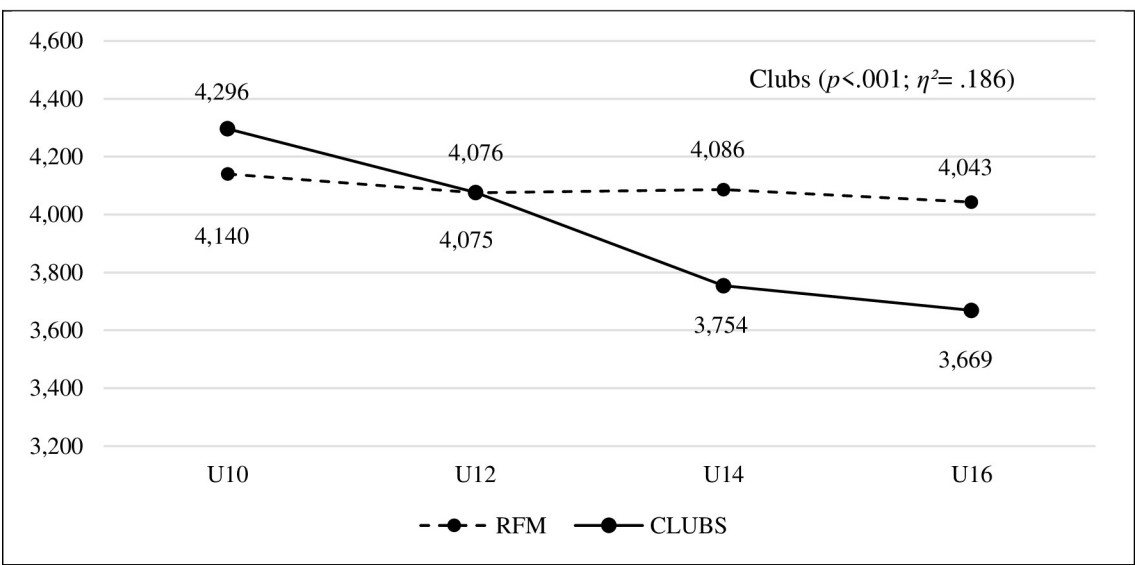

**Fig 1. Evolution of sportspersonship in the different categories of the FRM social ports schools and in the association football clubs.**

*hoc* analysis showed that club players had significant differences in all categories, with moderate and large effect sizes, except between the U14 and U16 categories, although there was a decrease in sportspersonship (Table 2).

When analysing the evolution of sportspersonship orientation according to each of aspect, ANOVA showed, for the FRM schools, that there was no difference in personal factors in the categories, although differences were found in social factors ($p = .006$) with a small effect size ($\eta^2 = .013$). The *post hoc* pairwise comparisons revealed that differences were only apparent between the U10 and U12 categories, with a small effect size.

There was a decrease in sportspersonship orientation in both dimensions in association football clubs. ANOVA identified differences in both personal factors ($p < .001$), with a large

**Table 2. Evolution of sportspersonship in the different categories of the FRM schools and the association football clubs.**

|  | Category | M±SD | Category | M±SD | p | d |
|---|---|---|---|---|---|---|
| **FRM** | **U10** | 4.140±.633 | **U12** | 4.075±.537 | - | - |
|  |  |  | **U14** | 4.086±.561 | - | - |
|  |  |  | **U16** | 4.043±.539 | - | - |
|  | **U12** | 4.075±.537 | **U14** | 4.086±.561 | - | - |
|  |  |  | **U16** | 4.043±.539 | - | - |
|  | **U14** | 4.086±.561 | **U16** | 4.043±.539 | - | - |
| **Clubs** | **U10** | 4.296±.450 | **U12** | 4.076±.506 | .022 | .46 |
|  |  |  | **U14** | 3.754±.539 | < .001 | 1.1 |
|  |  |  | **U16** | 3.669±.541 | < .001 | 1.3 |
|  | **U12** | 3.669±.541 | **U14** | 3.754±.539 | < .001 | .61 |
|  |  |  | **U16** | 3.669±.541 | < .001 | .78 |
|  | **U14** | 3.754±.539 | **U16** | 3.669±.541 | - | - |

Note. M = Mean; SD = Standard deviation; p = significance; d = Cohen's d.

Grouping variable: Categories.

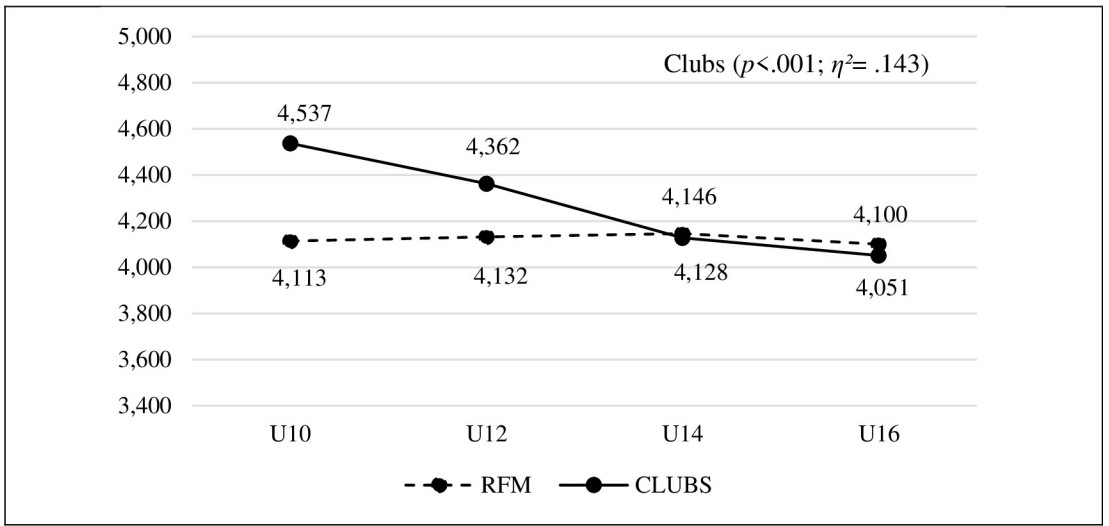

**Fig 2. Evolution of personal factors in the different categories in the FRM social sports schools and in the association football clubs.**

effect size ($\eta^2$ = .143), and social factors ($p < .001$), with a large effect size ($\eta^2$ = .152) (Figs 2 and 3). The *post hoc* pairwise comparisons found significant differences in almost all categories, with moderate to large effect sizes. Only in the U14 and U16 categories, no significant differences were found for both dimensions, though a certain decline in the data was still noticeable.

## Analysis of the differences in sportspersonship between the FRM schools and the association football clubs by category

The analysis of variance of two factors (2x4), Context (FRM *vs* Clubs) and Age Categories (U10, U12, U14, U16), in the personal factors dimension, detected that the interaction effect of

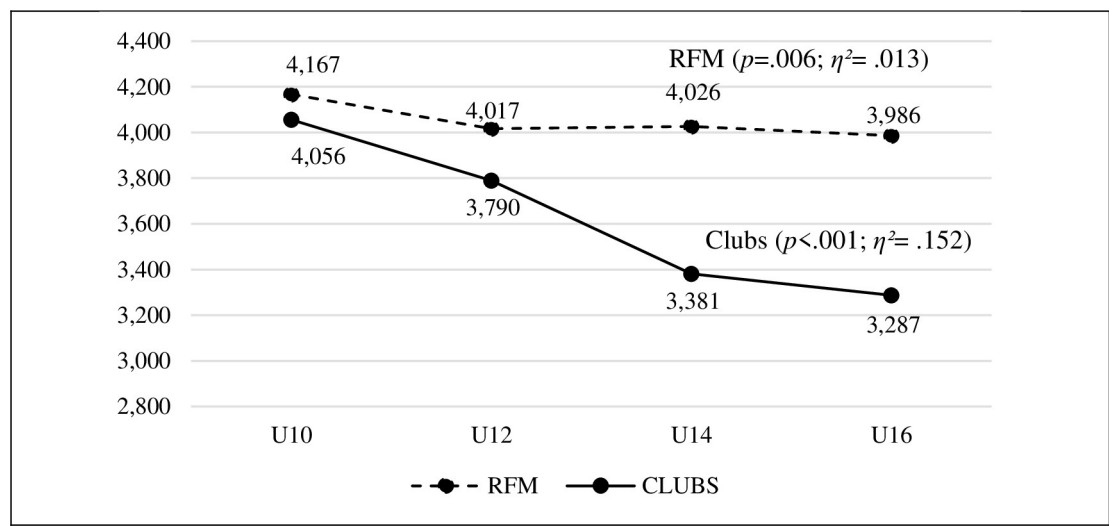

**Fig 3. Evolution of social factors in the different categories in the FRM social sports schools and in the association football clubs.**

**Table 3.**

|  |  | FRM | Clubs |  |
|---|---|---|---|---|
|  |  | M±SD | M±SD | *p* |
| **Personal factors** | **U10** | 4.114±.754 | 4.538±.458 | < .001 |
|  | **U12** | 4.133±.578 | 4.363±.412 | < .001 |
|  | **U14** | 4.147±.614 | 4.128±.523 | - |
|  | **U16** | 4.101±.531 | 4.052±.483 | - |
| **Social factors** | **U10** | 4.167±.671 | 4.056±.652 | - |
|  | **U12** | 4.017±.630 | 3.791±.733 | < .001 |
|  | **U14** | 4.026±.651 | 3.382±.717 | < .001 |
|  | **U16** | 3.988±.629 | 3.288±.781 | < .001 |

Note. M = Mean; SD = Standard deviation; p = significance; d = Cohen's d.

Grouping variable: Sport practiced.

the Context x Age Category factor is statistically significant ($F_{3,\,1501}$ = 11.806, *p*> .001, $\eta^2_p$ = .023), so it can be stated that the difference between the values can be explained by the interaction of both aspects.

When comparing both contexts according to the age categories, lower values appeared in the athletes of the WRF than in the athletes of the federated Clubs, so that there were statistically significant differences in the U10 and U12 age categories (U10: $F_{1,1501}$ = 50.684, *p* < .001, $\eta^2_p$ = .033; U12: $F_{1,1501}$ = 15.265, *p*< .001, $\eta^2_p$ = .010).

On the other hand, the personal factors related to sportsmanship remained practically constant throughout the different age categories in the FRM athletes, showing no significant differences ($F_{3,1501}$ = .211, *p* = .889, $\eta^2_p$ = .000), however, in the case of athletes from federated clubs, these factors decreased as the age Category increases, appreciating statistically significant differences ($F_{3,1501}$ = 18.344, *p*< .001, $\eta^2_p$ = .035).

Regarding the social factors dimension, after applying the analysis of variance of two factors (2x4), Context (FRM *vs* Clubs) and Age Categories (U10, U12, U14, U16), it can be seen that the interaction effect of the Context x Age Category factor is statistically significant ($F_{3,\,1501}$ = 15.237, *p*> .001, $\eta^2_p$ = .030), so it can be stated that the difference between the values can be explained by the interaction of both aspects.

When comparing both contexts according to the age categories, lower values appeared in the athletes of the federated clubs than in the athletes of the FRM, so that there were statistically significant differences in all age categories except U10, the differences being greater as the categories progressed (U12: $F_{1,1501}$ = 11. 281, *p*< .001, $\eta^2_p$ = .007; U14: $F_{1,1501}$ = 82.001, *p*< .001, $\eta^2_p$ = .052; U16: $F_{1,1501}$ = 54.253, *p*< .001, $\eta^2_p$ = .035).

Finally, the social factors related to sportsmanship, in the athletes of the FRM athletes showed a small decrease in values as the category advanced, ($F_{3,1501}$ = 3.830, *p* = .010, $\eta^2_p$ = .008). However, in the case of club athletes, a considerable decrease in the social factors of sportsmanship was observed, there were statistically significant differences ($F_{3,1501}$ = 36.738, *p*< .001, $\eta^2_p$ = .068) (Fig 3 and Table 3).

## Discussion

Previous research on sportspersonship has shown that its promotion in young players can prevent violent behaviour and foster a pro-social conduct [18,19]. Many authors defend the educational role of sport and consider that a deliberate and coherent planning according to this

goal is vital [20]. Thus, the aim of this research has been to analyse the evolution of sportspersonship orientation in the social sports schools of Fundación Real Madrid and in association football clubs.

When analysing the evolution of sportspersonship orientation in the different categories, it was noteworthy that the highest scores in sportspersonship in both contexts were obtained in the younger categories, which is consistent with the data obtained by Gómez-Marmol et al. [21]. The transition from childhood to adolescence brings young people to a new stage of moral development, acting in accordance with established norms, rather than out of fear of punishment or desire for reward. It is then that children begin to reconsider the validity of the actions taken and to make their own moral judgements [22]. However, it was relevant to see that the results of the players in the U10 category of FRM remained high in the rest of the categories. In this regard, Méndez-Giménez et al. [23] stress that sportspersonship in young athletes is linked to the fact that they all play at the same time and that they are polite to their opponents and the referee; Kavussanu and Spray [24], in their study on the influence of the context on the morality of young athletes, consider dialogue between players and coaches and the promotion of sportspersonship behaviour to be relevant, in agreement with the study by Wells et al. [8]. It is worth mentioning that these aspects are present in the socio-educational model of FRM [2]. Accordingly, focusing on the development of participation and enjoyment is related to positive fair play behaviour [25] and with the transmission of positive values [26]. Several authors [2,20] express that the use of sporting models specifically designed and planned to develop and promote values such as sportspersonship can have positive effects, as shown by numerous studies [23,27–35], even when used in a hybrid way [36].

The lack of sportspersonship in the early years of football training has been highlighted in numerous studies [18,37]. Accordingly, this research has observed a significant decrease of sportspersonship orientation in higher categories in association football clubs, which is in line with the findings of Papageorgiu et al. [38] and Wells et al. [34] and stresses the importance of working on values education in sports schools and clubs [39] throughout the entire training process. Many significant differences have been found, both in personal factors and in social factors with a large effect size. Players from clubs also show significant differences between categories, except between U14 and U16, although there was a decrease in sportspersonship orientation. On this subject, it should be noted that sports can contribute to a healthy physical and moral development in young people, although both positive and negative experiences can occur in sports, which can contribute to sports drop-out [40]. Thus, an atmosphere that is focused on "winning at all costs" can be key to influence children's participation in sports [8]. Most children tend to appreciate having fun as a reason and as a benefit of playing a sport, and they usually choose those activities [41]. Hence, one method to enhance the enjoyment of a youth sport experience may be to shift the focus of the programme to a greater emphasis on sportspersonship [8,34].

Data also show that personal factors ranked higher than social factors, which is in line with the research by Pelegrín [42], Gutiérrez and Pilsa [43] and Ortega et al. [44]. Furthermore, there were differences between the FRM social sports schools and the association football clubs in both dimensions. On the one hand, data on personal factors show significant differences, since the players of the association football clubs attached the greatest importance to these aspects. On the other, social factors ranked higher for the FRM players, in accordance with the results obtained by Gutiérrez [45] and Lamoneda et al. [39]. Therefore, the FRM social sports model can be considered to promote, to a large extent, aspects related to social ethics, such as respect for social conventions, rules, opponents and the referee, as well as helping others, which, however, is not in line with the findings of De Bofarull and Cusí [46], who found no differences in the promotion of sportspersonship between primary and secondary school students who practised only curricular sports and those who also practised extracurricular sports. These authors

conclude by stressing the need for very precise and deliberate interventions to promote sportspersonship [46], as it is done in the FRM social sports schools [2].

Regarding the evolution of sportspersonship orientation, taking into account the two different dimensions, in the FRM schools personal factors did not show different results throughout categories, although there were differences in the social factors. Regarding association football clubs, on the contrary, there was a significant decrease of sportspersonship orientation, both in personal and social factors. This might be related to young players having fun, as improving satisfaction and enjoyment has positive effects on their sporting behaviour [43,47]. Wells et al. [34] conclude their study by observing that, when an intentional programme to implement sportspersonship is carried out, not only does it increase significantly, but boys and girls have more fun. This highlights the need to decrease negative sporting behaviours in the game to enable young participants to have a more enjoyable experience [34], which is highly relevant in youth sports due to the high drop-out rate, with previous research showing that many children stop participating because they are no longer having fun [48]. Pavón and Moreno [49] agree when they state that, for young people, the most important reasons for practising a physical activity and sports are enjoyment and socialisation.

The results from clubs' players from categories U10 and U12 were higher than the data from the players from FRM in personal factors, but, in higher categories, no differences were found in this dimension, since the values of FRM remained homogeneous and high, while they had a significant decrease in federated clubs. On the other hand, regarding social factors, players from FRM obtained higher results compared to players from association football clubs in all categories. It is worth mentioning that pro-social behaviour refers to positive forms of social behaviour that are voluntary, not motivated by personal obligations, and that have positive social results [50]. Apart from that, one of the benefits of increasing sportspersonship in youth sports can be improving pro-social behaviour [34]. The environment of youth sports can be designed to promote pro-social behaviour by altering the overall competitive environment [51], as well as using referees to promote pro-social behaviour [52]. In this sense, referees can exemplify behaviour related to sportspersonship (e.g., helping a player to get up after falling), which can lead other players knowing and learning that behaving pro-socially is not only acceptable, but also expected [5]. All these aspects are part of the FRM sporting model and are implemented in its social sports football schools [2].

The main limitation of the study was that the data were collected through questionnaires, which may be biased to seek social acceptance [53]. In addition, most of the participants were boys, so the results of this study cannot be generalised to girls' pro-social behaviour. In terms of strengths, the research carried out is innovative because it is based on a comparison between two models in terms of the development of sportspersonship, as well as because of the large size of the sample.

Future research may be able analyse the effects on sportspersonship using sporting models such as the sports education model, the comprehensive teaching model, the personal and social responsibility model or other similar ones. It would also be interesting to determine whether boys and girls from the FRM social sports schools show a sporting attitude in their P.E. classes at school as well, or when they play in other sports schools, if applicable.

## Conclusions

The main conclusion of the research about the evolution of sportspersonship orientation shows that the FRM social sports model contributes to players maintaining high levels of sportspersonship over the years. On the contrary, in the association model, sportspersonship orientation decreases considerably as players move up through the categories.

Moreover, higher values are found in personal factors than in social factors in both contexts. However, players from the association football clubs rank personal factors more highly, while social factors are more highly appreciated by the FRM players. When comparing the two contexts in terms of the different dimensions, the personal dimension is more highly valued by players of the association football clubs in the lower U10 and U12 categories. On the contrary, athletes of the FRM social sports schools show higher values for social factors in all categories, especially in the U12, U14 and U16 categories.

In conclusion, it can be affirmed that the social sporting model of the FRM, in comparison with the association model, can contribute to a greater extent to maintaining high levels of sportspersonship orientation throughout the player's training process. On the contrary, the association model seems to lead to a decrease in sportspersonship orientation.

In practice, the study shows that the implementation of a sporting model focused on increasing positive sporting behaviour can help achieve the desired results. Hence, in order to increase sportspersonship among young people, sports institutions, clubs and organisations should encourage the implementation of educational programmes promoting values such as sportspersonship among young football players. To this end, a point of reference for them can be, among others, the characteristics of the FRM social sports model: recognition and encouragement of good behaviour, specific training for coaches to develop sportspersonship, educational competitions with regulations that favour sporting behaviour, and awareness raising among referees so that they facilitate and collaborate in the promotion of certain behaviour.

## Author Contributions

**Conceptualization:** Gema Ortega Vila, José Robles Rodríguez, Manuel Tomás Abad Robles, Francisco Javier Giménez Fuentes-Guerra.

**Data curation:** José Robles Rodríguez, Enrique Ortega Toro, Francisco Alarcón López.

**Formal analysis:** José Robles Rodríguez, Enrique Ortega Toro, Francisco Alarcón López.

**Investigation:** Gema Ortega Vila, José Robles Rodríguez, Manuel Tomás Abad Robles, Francisco Javier Giménez Fuentes-Guerra.

**Methodology:** Gema Ortega Vila, José Robles Rodríguez, Manuel Tomás Abad Robles.

**Resources:** Gema Ortega Vila, Manuel Tomás Abad Robles, Enrique Ortega Toro, Francisco Alarcón López, Francisco Javier Giménez Fuentes-Guerra.

**Supervision:** Gema Ortega Vila, José Robles Rodríguez, Manuel Tomás Abad Robles, Francisco Alarcón López, Francisco Javier Giménez Fuentes-Guerra.

**Writing – original draft:** José Robles Rodríguez, Manuel Tomás Abad Robles, Enrique Ortega Toro, Francisco Javier Giménez Fuentes-Guerra.

**Writing – review & editing:** José Robles Rodríguez, Manuel Tomás Abad Robles, Francisco Javier Giménez Fuentes-Guerra.

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
