## [Decision Letter · Decision Letter 0]

4 Sep 2024

PONE-D-24-06309Sportspersonship orientation in the Training Process of Young Football Players in Different Training ContextsPLOS ONE

Dear Dr. Abad Robles,

Thank you for submitting your manuscript to PLOS ONE. After careful consideration, we feel that it has merit but does not fully meet PLOS ONE’s publication criteria as it currently stands. Therefore, we invite you to submit a revised version of the manuscript that addresses the points raised during the review process.

**ACADEMIC EDITOR: **Dear authors, reviewer 3 requested a minor revision of the manuscript, please respond to his comments and send the revised version of the manuscript

We look forward to receiving your revised manuscript.

Kind regards,

Jovan Gardasevic

Academic Editor

PLOS ONE

3. Please declare the competing interests in the Manuscript in association with Real Madrid Foundation

“Real Madrid Foundation.”

5. We note that your Data Availability Statement is currently as follows: [All relevant data are within the manuscript and its Supporting Information files.]

Reviewers' comments:

Reviewer's Responses to Questions

**Comments to the Author**

1. Is the manuscript technically sound, and do the data support the conclusions?

Reviewer #1: Partly

Reviewer #2: Yes

2. Has the statistical analysis been performed appropriately and rigorously? 

Reviewer #1: No

Reviewer #2: Yes

3. Have the authors made all data underlying the findings in their manuscript fully available?

Reviewer #1: Yes

Reviewer #2: Yes

4. Is the manuscript presented in an intelligible fashion and written in standard English?

Reviewer #1: Yes

Reviewer #2: Yes

5. Review Comments to the Author

Reviewer #1: The introduction of the article is too extensive. You need to shorten it.

The goal of the article should be written in the past tense.

The statistical analysis was done incorrectly, therefore the results presented in this study are not correct.

Reviewer #2: very well put up work. from the title of the topic, through the set goal and methodology of work, presented results and discussion. Apart from the fact that works on this topic are very rare, the authors used a very quick quality system to monitor the variables important for the selected topic. the authors discussed a large number of bibliographic units. the sample is very representative. the results are shown.

very nice. I sincerely recommend this work for publication.

6. PLOS authors have the option to publish the peer review history of their article (what does this mean?). If published, this will include your full peer review and any attached files.

Reviewer #1: No

Reviewer #2: No

---

## [Author Response · Author response to Decision Letter 0]

26 Sep 2024

REBUTTAL LETTER

Manuscript PONE-D-24-06309

Sportspersonship orientation in the Training Process of Young Football Players in Different Training Contexts

PLOS ONE

Reviewer 1’s comments and suggestions for authors and details of the revisions and responses

1.- The introduction of the article is too extensive. You need to shorten it.

We have made the change indicated.

2.- The goal of the article should be written in the past tense.

We have made the change indicated:

That is why the aim of this study was to analyse the evolution of sportspersonship orientation in the social sports schools of Fundación Real Madrid (FRM) and in association football clubs.

3.- The statistical analysis was done incorrectly, therefore the results presented in this study are not correct.

Taking into account the considerations of the reviewer, we consider that the statistical analysis carried out is correct, as this analysis responds to the objectives of the research. However, we have considered the reviewer's comment and made some changes in order to carry out a more complete statistical mathematical approach since we have carried out a multifactorial analysis.

Specifically, a basic descriptive analysis of central trend and dispersion of each dimension was performed. To determine whether there were differences between the two contexts (FRM vs. federated Clubs), the T-Student contrast test was used for independent samples. In cases where the data showed significant differences (p .05), effect size was calculated using standardized mean differences (size of Cohen effect). To examine whether there was an interaction effect between Context (FRM vs. Clubs) and Age Categories (U10, U12, U14, U16), a two-factor ANOVA (2x4) was performed. An analysis of estimated marginal means was performed; simple main effects were compared by post hoc testing with Bonferroni's confidence interval adjustment.

THANK YOU for your comments and suggestions.

---

## [Decision Letter · Decision Letter 1]

2 Oct 2024

Sportspersonship orientation in the Training Process of Young Football Players in Different Training Contexts

PONE-D-24-06309R1

Dear Dr. Manuel Tomás Abad Robles,

We’re pleased to inform you that your manuscript has been judged scientifically suitable for publication and will be formally accepted for publication once it meets all outstanding technical requirements.

Kind regards,

Jovan Gardasevic

Academic Editor

PLOS ONE

Additional Editor Comments (optional):

Reviewers' comments:

Reviewer's Responses to Questions

**Comments to the Author**

1. If the authors have adequately addressed your comments raised in a previous round of review and you feel that this manuscript is now acceptable for publication, you may indicate that here to bypass the “Comments to the Author” section, enter your conflict of interest statement in the “Confidential to Editor” section, and submit your "Accept" recommendation.

Reviewer #3: All comments have been addressed

2. Is the manuscript technically sound, and do the data support the conclusions?

Reviewer #3: (No Response)

3. Has the statistical analysis been performed appropriately and rigorously? 

Reviewer #3: (No Response)

4. Have the authors made all data underlying the findings in their manuscript fully available?

Reviewer #3: (No Response)

5. Is the manuscript presented in an intelligible fashion and written in standard English?

Reviewer #3: (No Response)

6. Review Comments to the Author

Reviewer #3: (No Response)

7. PLOS authors have the option to publish the peer review history of their article (what does this mean?). If published, this will include your full peer review and any attached files.

Reviewer #3: No

---

## [Editor Report · Acceptance letter]

8 Oct 2024

PONE-D-24-06309R1 

PLOS ONE

Dear Dr. Abad Robles, 

I'm pleased to inform you that your manuscript has been deemed suitable for publication in PLOS ONE. Congratulations! Your manuscript is now being handed over to our production team.

Kind regards, 

on behalf of

Dr. Jovan Gardasevic 

Academic Editor

PLOS ONE